# Short-Term Effects of the Repeated Exposure to Trip-like Perturbations on Inter-Segment Coordination during Walking: An UCM Analysis

**Vito Monaco** [1,2,3,*], **Clara Zabban** [1,4] **and Tamon Miyake** [5]

1    The BioRobotics Institute, Scuola Superiore Sant'Anna, 56127 Pisa, Italy; clara.zabban@santannapisa.it
2    Department of Excellence in Robotics and AI, Scuola Superiore Sant'Anna, 56127 Pisa, Italy
3    IRCCS Fondazione Don Carlo Gnocchi, 20148 Milan, Italy
4    Scuola di Ingegneria, Università di Pisa, 56126 Pisa, Italy
5    Faculty of Science and Engineering, Waseda University, Tokyo 169-8555, Japan; tamonmiyake@aoni.waseda.jp
*    Correspondence: vito.monaco@santannapisa.it

**Abstract:** The minimum toe clearance (MTC) results from the coordination of all bilateral lower limb body segments, i.e., a redundant kinematic chain. We tested the hypothesis that repeated exposure to trip-like perturbations induces a more effective covariation of limb segments during steady walking, in accordance with the uncontrolled manifold (UCM) theory, to minimize the MTC across strides. Twelve healthy young adults (mean age 26.2 ± 3.3 years) were enrolled. The experimental protocol consisted of three identical trials, each involving three phases carried outin succession: steady walking (baseline), managing trip-like perturbations, and steady walking (post-perturbation). Lower limb kinematics collected during both steady walking phases wereanalyzed in the framework of the UCM theory to test the hypothesis that the reduced MTC variability following the perturbation can occur, in conjunction with more effective organization of the redundant lower limb segments. Results revealed that, after the perturbation, the synergy underlying lower limb coordination becomes stronger. Accordingly, the short-term effects of the repeated exposure to perturbations modify the organization of the redundant lower limb-related movements. In addition, results confirm that the UCM theory is a promising tool for exploring the effectiveness of interventions aimed at purposely modifying motor behaviors.

**Keywords:** minimum toe clearance; walking; uncontrolled manifold; repeated exposure; perturbation; tripping

## 1. Introduction

The minimum toe clearance (MTC) is the distance between toe and ground, as assessed when the time course of the toe's vertical displacement reaches the relative minimum during the mid-swing phase of a gait cycle [1]. At this critical time, if the trajectory of the swinging foot is abruptly interrupted by an obstacle (i.e., tripping), the overall dynamic is challenged and the consequent lack of balance can result in a fall. The MTC is purported to be a measure of the risk of tripping [1–4]. Accordingly, understanding how humans control MTC is of paramount importance to designing suitable strategies to decrease fall risk [2,3].

Earlier literature demonstrated that MTC average and variability across strides reflect the individual's attitude to controlling the toe clearance while walking. In more detail, a lower MTC average results from an increased cognitive workload [5,6], while reduced MTC variability documents a more precise neuromuscular control of the toe clearance [3,7,8]. In one of our recent works, we extended previous findings to reveal that repeated exposure to a series of trip-like perturbations modifies MTC average and variability towards a more conservative (i.e., lower MTC average) and more precise (i.e., lower MTC variability) neuro-muscular control strategy [9].

In this study, we further investigated the short-term effects of repeated exposure to trip-like perturbations on steady walking, to provide a more in-depth analysis of inter-segmental coordination. From a kinematic viewpoint, MTC is the result of suitable coordination of all lower limb body segments, from the foot of the trailing limb to the contralateral one. Noticeably, this kinematic chain is redundant because the number of available degrees of freedom (DoFs), i.e., the orientation of all body segments, is far larger than the controlled variable (i.e., the vertical distance between toe and ground). Therefore, different configurations of lower limb segments can result in the same MTC.

According to this evidence, we tested the hypothesis that amore precise control of the toe clearance (reduced MTC variability) after the perturbation occurs in conjunction with a more effective covariation of the redundant lower limb segments. To test this hypothesis, we adopted the uncontrolled manifold (UCM) theory [10,11]. This approach relies on the analytical relationship between the redundant DoFs of the musculoskeletal system (i.e., the orientation of lower limb segments), also named *elemental variables*, and the *task performance*, representing the putative controlled variable (the MTC; see Section 2.3 for further details). We hence tested the hypothesis that the variability across repetitions of *elemental variables* is mostly limited to a set of solutions that minimises the homologous variability of the *task performance*.

## 2. Materials and Methods

The dataset used in this study is a subset of that collected in our previous work [9], and refers to 12 of 14 recruited subjects. Data concerning 2 subjects were discarded because the 3D track of their anterior-iliac spines were missed for long periods (i.e., tens of seconds). Therefore, it was not possible to accurately estimate hip joint position.

The following subsections will briefly recapitulate the procedures and methods already described elsewhere [9,12].

### 2.1. Participants, Experimental Setup, and Protocol

Twelve healthy young adults (7 M/5 F; mean $\pm$ standard deviation; age: $26.2 \pm 3.3$; height: $1.72 \pm 0.09$; mass: $64.4 \pm 8.7$ kg) were enrolled inthis study. They were asked to walk on a treadmill at their preferred speed ($0.90 \pm 0.08$ m/s), either in steady conditions or while managing unexpected trip-like perturbations.

The experimental setup consisted of a treadmill equipped with a mechatronic platform, hosting two nylon ropes connecting both feet with its main frame through a couple of compliant springs (Figure 1) [9,13]. The rope connecting the right foot ran through a cam-based braking mechanism and could be stopped to interrupt the forward movement of the right foot during the mid-swing phase, thus emulating a trip. To minimize inter-subject variability, each perturbation was triggered at the heel strike of the contralateral foot.

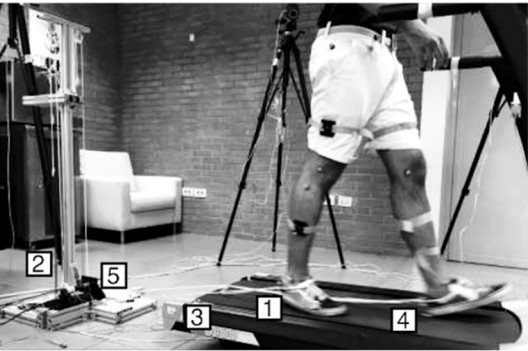

**Figure 1.** This picture provides an overview of the experimental setup. Labelled components are: 1. Foot being perturbed; 2. Perturbing platform; 3. Treadmill; 4. Unperturbed foot; 5 Cam-braking mechanism.

The experimental protocol consisted of three identical trials (A, B, and C) each accounting for three sub-phases occurringin succession: i. steady walking on the treadmill for about 2 min (baseline); ii. managing 30 unexpected trip-like perturbations delivered at random every 3–20 strides; iii. steady walking for about 2 min (post-perturbation). To prevent proactive adjustments due to external cues, participants listened to music via headphones and did not know when the perturbations were to be delivered. For safety purposes, the treadmill was equipped with handrails, which participants were instructed to grasp only in cases of unrecoverable lack of balance. Otherwise, they were not allowed to use the handrails during the experimental session.

The study protocol was approved by the Local Ethics Committee. Enrolled subjects provided written, informed consent before starting experimental sessions.

### 2.2. Data Collection and Processing

The 3D trajectory of 25 lower limb body landmarks was recorded at 100 Hz by using a six-camera, motion capture system (Vicon 512 Bonita 10 Motion Analysis System, Oxford, UK; Figure 2). Collected kinematics was pre-processed according to the following pipeline: i. gap filling, if any; ii. zero-lag, low-pass filtering (Butterworth, 4th order) with a cut-off at 10 Hz.

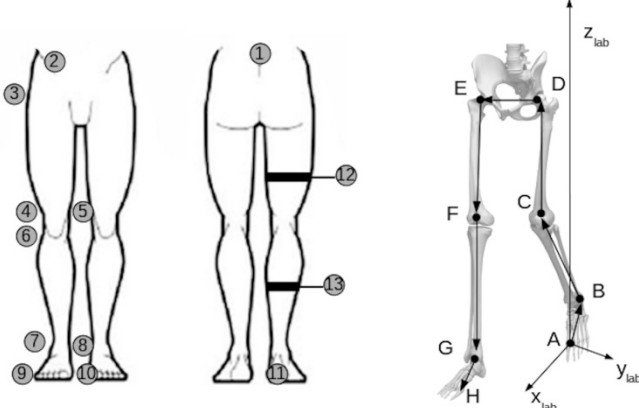

**Figure 2.** On the left panel, half (right side) body marker set is reported as follows: 1. middle point between posterior iliac spines; 2. anterior iliac spine; 3. prominence of the greater trochanter external surface; 4. lateral epicondyle of the femur; 5. medial epicondyle of the femur; 6. head of the fibula; 7. lateral malleolus; 8. and medial malleolus; 9. 5th metatarsal head; 10. 1st metatarsal head; 11. calcaneus; 12. marker rigidly attached to a wand over midfemur; 13. marker rigidly attached to a wand over midshaft of the tibia. On the right side, the 7-segments kinematical model is reported. Axes $x_{lab}$, $y_{lab}$, and $z_{lab}$ refer to the lab-related reference frame.

The 3D trajectory of the markers on the right foot were used to identify the following time events: the heel strike and the toe off, as maximum and minimum elevation of the limb axis, in compliance with previous literature [14,15]; the time instant during the swing phase corresponding to the local minimum of 1st metatarsal head ($t_{MTC}$). For each subject in each trial (A, B, and C) and each sub-phase (baseline and post-perturbation), the mean and the standard deviation of the MTC across collected strides ($MTC_{av}$ and $MTC_{SD}$) were computed.

Kinematics as assessed at $t_{MTC}$ were also used to implement the UCM analysis.

### 2.3. UCM Implementation

The UCM approach aims at testing the hypothesis that available DoFs, also referred to as *elemental variables*, covary across repetitions to minimize the variability of the controlled variable, namely *task performance* [10,11,16]. In the framework of this study, we used segmental angles rather than joint angles as *elemental variables* according to the evidence that segmental angles covary during numerous walking-related tasks [17–21] while joint angles

do not [22], under the assumption of planar approximation of locomotion. To test the hypothesis that the redundant DoFs represent an adaptable source of solutions to ensure a stable MTC (i.e., *task performance*) across strides, the inter-stride variability of *elemental variables* is split in two components: i. the former defines a domain of segmental orientations leading to the same MTC value across strides; ii. the latter defines an orthogonal domain accounting for all solutions that do modify the MTC across strides. When the variability of *elemental variables* is mostly confined to the former domain, namely the uncontrolled manifold, it is possible to conclude that the orientation of lower limb segments is functionally organized across strides to minimize the MTC inter-stride variability (i.e., $MTC_{SD}$). Noticeably, the average number of strides collected across subjects and trials was $25 \pm 2$ in accordance with earlier findings [23].

The algorithm adopted to implement the UCM analysis resembles that reported in our previous works [12,24]. Briefly, a 3D kinematics model of lower limbs, accounting for seven body segments (feet, shanks, thighs, and pelvis) and six spherical joints (hips, knees, and ankles) was developed, as shown in Figure 2. The location of each joint was estimated as follows: the ankle joint coincided with the middle point between medial and lateral malleolus; the knee joint coincided with the middle point between medial and lateral epicondyle of the femur; the hip joint coincided with the acetabulum whose location was computed as described elsewhere [25], based on the markers on anterior iliac spines and middle point between posterior iliac spines. The orientation of each body segment was estimated in terms of azimuth and elevation angles (spherical coordinates) from the *x*-axis of the lab-related reference frame. The trigonometric model relating *elemental variables* (elevation and azimuth angles of lower limb body segments) and *task performance*, i.e., the MTC of the right foot, was developed according to the Equation (1):

$$MTC = AH_z = \sum_{m=1}^{7} L_m \cdot cos(\varepsilon_m) \cdot sin(\alpha_m) \tag{1}$$

where:

- $AH_z$ coincides with the MTC;
- $L_m$, $\varepsilon_m$, and $\alpha_m$ refer to length, elevation, and azimuth angles of the *m* body segment;
- $m = 1, 2, \ldots, 7$ refers to the ordered series of body segments, from the left foot to the right one.

The Jacobian matrix (J) of the function in Equation (1), relating small changes in the elemental variables (i.e., $\varepsilon_1, \alpha_1, \ldots, \varepsilon_m$, and $\alpha_m$) to task performance (i.e., MTC), was hence computed. J was calculated around the mean configuration of segmental angles across strides (i.e., $\overline{\varepsilon_1}, \overline{\alpha_1}, \ldots, \overline{\varepsilon_m}$, and $\overline{\alpha_m}$). Then, its null space (N(J)) was estimated as a linear approximation of the UCM, i.e., when the deviation of segmental angles from the mean configuration is confined into the N(J), the MTC does not change, and viceversa. After that, the deviation of *elemental variables* from their mean values across strides was calculated and projected onto and orthogonal to the UCM (namely, $DV_{UCM}$ and $DV_{ORT}$, respectively). Finally, the variances across strides of these projections onto and orthogonal to the UCM, (namely $\sigma^2_{UCM}$ and $\sigma^2_{ORT}$) were computed across strides and normalised per degree of freedom of each subspace. The ratio between variance components was computed as a synthetic synergy index (the greater the *Ratio*, the stronger the synergy of the underlying lower limb coordination).

### 2.4. Statistical Analysis

Mean and standard deviation were used as descriptive statistics to refer to the central tendency and dispersion of all independent variables (i.e., $MTC_{av}$, $MTC_{SD}$, $\sigma^2_{UCM}$, $\sigma^2_{ORT}$ and *Ratio*). A two-way repeated measures Analysis of Variance (ANOVA) was implemented to investigate the main and interaction effects of factors trials (three levels: A, B, and C) and sub-phase (two levels: baseline and post-perturbation) on the outcome variables. The

one-sample t-test was carried out to test the hypothesis that the mean of the *Ratio* is equal to 1, for each trial and each sub-phase.

Significance for all statistical tests was set at $p < 0.05$. Data analysis was carried out using Matlab R2020a (The MathWorks, Inc., Natick, MA, USA).

## 3. Results

Table 1 reports both the descriptive statistics and outcomes of the statistical analysis for all independent variables.

**Table 1.** Independent variables (mean $\pm$ standard deviation) as assessed across experimental conditions and outcome of the statistical analysis. Acronyms $p_T$; $p_{Sp}$, $p_{int}$ represent the outcome of the statistical analysis in terms of *p*-values with respect to factors trial (three levels: A, B, and C), sub-phase (two levels: baseline and post-perturbation), and their interaction, respectively. *p*-values are reported in bold when significant ($p < 0.05$).

| Variable | Trial A | | Trial B | | Trial C | | *p*-Values |
|---|---|---|---|---|---|---|---|
| | **Baseline** | **Post-pert.** | **Baseline** | **Post-pert.** | **Baseline** | **Post-pert.** | |
| $MTC_{av}$ (mm) | $34.8 \pm 9.1$ | $31.1 \pm 10.2$ | $33.1 \pm 10.0$ | $30.9 \pm 8.9$ | $32.6 \pm 8.3$ | $31.1 \pm 8.6$ | $p_T = 0.966$ **$p_{Sp} = 0.001$** $p_{int} = 0.197$ |
| $MTC_{SD}$ (mm) | $3.4 \pm 1.0$ | $3.1 \pm 1.0$ | $3.3 \pm 1.4$ | $3.0 \pm 0.7$ | $3.4 \pm 1.1$ | $3.1 \pm 0.9$ | $p_T = 0.602$ **$p_{Sp} = 0.012$** $p_{int} = 0.795$ |
| $\sigma^2_{UCM}$ (rad$^2$) | $0.16 \pm 0.24$ | $0.19 \pm 0.18$ | $0.11 \pm 0.14$ | $0.16 \pm 0.17$ | $0.13 \pm 0.18$ | $0.18 \pm 0.18$ | $p_T = 0.841$ $p_{Sp} = 0.159$ $p_{int} = 0.972$ |
| $10^3 \times \sigma^2_{ORT}$ (rad$^2$) | $0.08 \pm 0.04$ | $0.11 \pm 0.08$ | $0.14 \pm 0.22$ | $0.13 \pm 0.15$ | $0.11 \pm 0.15$ | $0.09 \pm 0.06$ | $p_T = 0.498$ $p_{Sp} = 0.712$ $p_{int} = 0.769$ |
| $10^{-3} \times Ratio$ (adim) | $1.58 \pm 2.06$ | $2.92 \pm 3.73$ | $1.56 \pm 2.45$ | $2.15 \pm 3.23$ | $1.65 \pm 2.62$ | $2.86 \pm 4.14$ | $p_T = 0.881$ **$p_{Sp} = 0.015$** $p_{int} = 0.926$ |

The results confirmed that both $MTC_{av}$ and $MTC_{SD}$ significantly ($p_{Sp} < 0.05$) decreased after the repeated exposure to perturbation [9]. With regards the outcome of the UCM analysis, both before and after the perturbation, $\sigma^2_{UCM} > \sigma^2_{ORT}$ (i.e., *Ratio* > 1; $p < 0.05$). In addition, due to the perturbation, $\sigma^2_{UCM}$ increased, albeit without statistical significance ($p_{Sp} = 0.159$), whereas $\sigma^2_{ORT}$ remained almost constant. These variations led to a significant growth of the *Ratio* ($p_{Sp} = 0.015$). Neither the effect of the factor trials nor that of the interaction between factors trials and sub-phases were noticed for all independent variables (i.e., $p_T > 0.05$ and $p_{int} > 0.05$ for all comparisons).

## 4. Discussion

This study investigated the short-term effects of the repeated exposure to trip-like perturbations on inter-segmental coordination during steady walking, as a follow-up analysis of a dataset collected for one of our recent works [9]. Here, we tested the hypothesis that, due to perturbation, a more precise control of the toe clearance (reduced $MTC_{SD}$) can occur, in conjunction with more effective organisation of the redundant lower limb segments, in accordance with the UCM theory. The results confirmed that, after the perturbation, the synergic covariation of segmental orientation further stabilises the MTC variability across strides.

As an initial result, this study demonstrates that lower limb intersegmental coordination while walking is functionally structured to minimise the inter-stride MTC variability in accordance with the UCM theory (*Ratio* > 1). Notably, previous authors have dealt with different performance variables related to the swinging foot, such as mediolateral

footpath trajectory [26–29], or whole 3D footpath [12,24], and reported slightly discordant outcomes [12]. Our findings, in conjunction with previous ones [12,24,26–29], suggest that the central nervous system may aim at individually and flexibly stabilising the inter-stride variability of footpath components (e.g., mediolateral, vertical) during different periods of the swing phase. This hypothesis deserves to be further investigated to gain a more general understanding concerning the organisation of inter-stride variability of lower limb body segments during the swing phase.

The main result is related to the decrease in $MTC_{SD}$ and suggests that the strength of the kinematic synergy underlying MTC stabilisation across strides further increases after the perturbation (se *Ratio* in Table 1). In our previous work, we argued that the post-perturbation reduction in$MTC_{av}$ and, especially, $MTC_{SD}$ would reflect a volitional strategy to achieve a more effective compensatory step in case of forthcoming perturbations [9]. The results reported in this study further extend this conclusion, suggesting that the participants could proactively modify the coordination of all segments belonging to both the trailing and the leading limbs. This evidence confirms that the UCM theory is a promising tool for exploring the effectiveness of interventions aimed at purposely modifying motor behaviours, as also documented in earlier studies [12,24,30]. In addition, they suggest that the short-term effects of the repeated exposure to perturbations may involve a modified interaction between the bilateral neural structures leading lower limb-related movements. Further in-depth investigations are required to explore the relationship between the outcome of the UCM analysis and the neural control of bipedal locomotion.

*Limits of the Study*

One of the main limits of this study consisted of the small number of enrolled subjects (i.e., twelve healthy young adults) resulting in a limited strength of the statistical findings. However, it should be noticed that the outcomes of the statistical analysis that reached the significance (Table 1) were typically far from the threshold (i.e., $p = 0.05$). Accordingly, we can confidently conclude that results can be considered robust enough. Another limit is that findings reported in this study are related to young healthy adults and cannot be generalized to other groups of individuals, such as elderly people or people affected by neuro-muscular diseases. In this respect, we believe that further and specific analysis are required to investigate the effects of the proposed intervention on people belonging to other populations.

## 5. Conclusions

This study shows that lower limb inter-segments coordination while walking is functionally structured to minimise the inter-stride variability of the MTC in accordance with the uncontrolled manifold (UCM) theory. In addition, after repeated exposure to trip-like perturbations, the intersegmental coordination underlying the MTC stabilisation across strides becomes stronger, as assessed by the ration between variance components. Future research in motor control theory based on the UCM is expected to investigate more extensively the relevance of the single components of the 3D foot trajectory during the swing phase, and the relationship between highlighted coordinative strategy and neural control of locomotion.

**Author Contributions:** Conceptualization, V.M.; methodology, V.M. and T.M.; software, V.M. and C.Z.; formal analysis, V.M. and C.Z.; data curation, V.M.; writing—original draft preparation, V.M.; writing—review and editing, V.M., C.Z., and T.M.; supervision, V.M. All authors have read and agreed to the published version of the manuscript.

**Funding:** This work was supported by: the "Graduate Program for Embodiment Informatics" of the Ministry of Education, Culture, Sports, Science and Technology (MEXT) of Japan; the EU Commission through the H2020 project CYBERLEGs Plus (Grant Agreement no. 731931); the Italian National Institute for Insurance against accidents at work (INAIL) within the MOTU project (PPR-AI 1-2); Grants-in-Aid for Scientific Research of the Japan Society for the Promotion of Science (19J14599); institutional funds of the Scuola Superiore Sant'Anna, Pisa, Italy.

**Institutional Review Board Statement:** The study was conducted according to the guidelines of the Declaration of Helsinki, and approved by the Ethics Committee Comitato Etico Regionale per la Sperimentazione Clinica nella Regione Toscana, Sez. Area Vasta Centro (protocol code: MOTU ATP; date of approval: 21 July 2020).

**Informed Consent Statement:** Informed consent was obtained from all subjects involved in the study.

**Data Availability Statement:** The data presented in this study are available on request from the corresponding author.

**Acknowledgments:** The authors would like to thank Federica Aprigliano and Chiara Fanciullacci for their valuable support and precious assistance.

**Conflicts of Interest:** The authors declare no conflict of interest.

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
