# Peer review of "Short-Term Effects of the Repeated Exposure to Trip-like Perturbations on Inter-Segment Coordination during Walking: An UCM Analysis"

_applsci, doi:10.3390/app11209663_

Round 1
Reviewer 1 Report
The submitted manuscript is not original as a similar article was published by Miyake et al. in 2021 (Miyake, T., et al., Repeated exposure to tripping like perturbations elicits more precise control and lower toe clearance of the swinging foot during steady walking. Human Movement Science, 2021. 76: p. 102775.). Both manuscripts used the identical methods to answer the same research questions, even though the author used an UCM analysis. Large parts of both manuscripts are identical. The submitted manuscript does not fulfill the first of the global issues that are recommended for consideration by the editor (i.e. "Is the research original?").
Author Response
Please, see attached document

Reviewer 2 Report
I read the manuscript "Short term effects of the repeated exposure to tripping like perturbations on the inter-segments coordination during walking: an UCM analysis" by Monaco et al. with interest, and albeit the study was a post-hoc analysis, I found the paper has its merit using UCM theory for responses to perturbations. However I suggest authors to address potential points below.
1. The analysis is generally descriptive and as authors addressed, some presentations are quite similar to the recent previous work (Ref #9). I suggest authors comment in the discussion section that the study is a post-hoc analysis of existing dataset.
2. I suggest authors to add a 'limitation' paragraph to address following weaknesses. The study was performed only in small number of young healthy adults, and findings might not be generalizable to other population. Also, the study simulated tripping like perturbations, and may not reflect the real world setting.
Author Response
Please, see attached document

Reviewer 3 Report
Point 1: I would like to discuss some issues that may need to be addressed in the manuscript. What is the procedure for randomly assigning study subjects to each trials (A, B, and C) ? Please Include this procedure in "design" section of the method.
Author Response
Please, see attached document

Reviewer 4 Report
Overall
- Overall, this is an interesting, succinct, technical-style report that aims to analyze a subset of data from a previous gait study by means of the uncontrolled manifold approach
- All sections are clear and straightforward in their presentation
- UCM is a heavy analysis method that may be new to certain readers. The authors do a good job of explaining the basics in their methods section. They also provide some contextual references that readers could seek out
- This reviewer does not see any glaring omissions in methodology or discussion that needs significant revision
- Further discussion on the previously reported, unexpected decrease in MTC versus increase (as previously hypothesized) might be warranted
- Further discussion on the future implications of this work for the future of gait analysis, particularly tripping and balance, may also strengthen this work (this is just a suggestion)
Abstract
- Straightforward and clear
Introduction
- Throughout all sections, there are unnecessary uses of the word “the” and I would recommend some copy-editing before publication
- Line 33 – “trajectory” instead of time course?
- Line 36 – dynamics “are” challenged
- Line 39 – decrease fall risk
- Line 56 – I’m not sure this sentence makes sense? Maybe – “We tested the hypothesis that more precise control of toe clearance (reduced MTC variability) after perturbation results in more effective covariation of redundant lower limb segments.”
Methods
- Line 74 – mean ± standard deviation is self explanatory, I’m not sure you need the explicit definition within the parentheses
- Line 97 – were there any other safety measures taken here? Were the subjects suspended in some way such that if they really fell, they would not crash onto the treadmill? Added safety precautions seem important to note here
- Line 106 – no need for “this is a figure”
- Line 109 – you have a comma instead of a period following #6
- Lines 125-168 – the uncontrolled manifold theory indeed seems appropriate here and this is a straightforward, succinct explanation of your application of UCM. A full explanation of UCM and its theory/application is available in the references and beyond the scope of this short article. The sentence in line 137-140 is the key – the ratio of variability in the uncontrolled vs. controlled is your main outcome and the main hypothesis being tested through UCM
- Line 141 – does it “likely” resemble or does it resemble?
- Line 178 – sometimes useful to include a version number here, like r2020b
Results
- Straightforward reporting of results
Discussion
- It is discussed in your previous, referenced work, but it might be worth adding a short discussion paragraph noting that the DECREASE in MTC was actually counter to your original hypothesis. You would expect, and some previous research has shown, that MTC actually increases due to the threat of tripping to allow more time for compensation. The previous work [9] does a nice job explaining this away and it might be worth including some of those arguments
- Line 205 – if this is indeed the first study to use UCM for this, you can keep it, but I would suggest softening “this is the first study…” phrasing. Without doing a complete literature search, other studies have shown similar results using methods like TNC and I would guess the space has been explored, maybe in non-identical ways, with UCM (https://www.sciencedirect.com/science/article/pii/S096663622030196X)
- Line 216 – if this is indeed the main results it is mainly related to the decrease in
Conclusions
- Line 233 – the sentence beginning on this line needs reworded
Author Response
Please, see attached document

Round 2
Reviewer 1 Report
L111 7. medial malleolus and 8. lateral malleolus should exchange. I was wrong marked.
L121 How to determine heel strike and toe off should be elaborated.
L124 How many strides were analyzed for each trial?
L148 What is the difference between the global reference frame and lab-related reference frame should be elaborated.
L227 …also documented by [12, 22, 27]. [12, 22, 27] should be replaced by authors’ names.
